# Physico-Chemical, Microbiological and Sensory Evaluation of Ready-to-Use Vegetable Pâté Added with Olive Leaf Extract

**DOI:** 10.3390/foods8040138

**Published:** 2019-04-23

**Authors:** Graziana Difonzo, Giacomo Squeo, Maria Calasso, Antonella Pasqualone, Francesco Caponio

**Affiliations:** Department of Soil, Plant and Food Science (DISSPA), University of Bari Aldo Moro, Via Amendola, 165/a, I-70126 Bari, Italy; graziana.difonzo@uniba.it (G.D.); giacomo.squeo@uniba.it (G.S.); maria.calasso@uniba.it (M.C.); antonella.pasqualone@uniba.it (A.P.)

**Keywords:** olive leaves, waste, bioactive compounds, shelf-life, antimicrobial properties, acceptability

## Abstract

The shelf-life extension implicates the reduction of food waste. Plant polyphenols can have a crucial role in the shelf-life extension of foods. Olive leaf extract (OLE) is rich in phenolic compounds such as oleuropein, which is well-known for its antioxidant properties. Physico-chemical, microbiological and sensory aspects of non-thermally stabilized olive-based pâté fortified with OLE at concentrations of 0.5 (EX0.5) and 1 mg kg^−1^ (EX1) were investigated. These samples were compared with olive-based pâté fortified with the synthetic antioxidant BHT (butylated hydroxytoluene) and with a control sample (CTR) without antioxidants. No sensory defects were perceived in all samples, even if a more intense typical olive flavour was perceived in samples containing OLE compared to those containing BHT and CTR. This result was confirmed by significantly higher levels of 2-methylbutanal and 3-methylbutanal in samples containing OLE compared to CTR and BHT. Moreover, the main microbial groups registered a significant loss of 0.5–1 logarithmic cycles in samples containing OLE, especially in EX1. The results of the present study indicate the potentiality of using OLE as natural preservatives in non-thermally stabilized olive-based pâté, since some spoilage-related microbial groups were negatively affected by the addition of OLE at the highest concentration.

## 1. Introduction

Shelf-life is defined as a finite length of time, after manufacture and packaging, during which the food product retains at a required level of quality acceptable for consumption. Shelf-life is a function of time, environmental factors and susceptibility of product to quality changes [1]. Physico-chemical and biological changes that take place in the food chain are involved in product deterioration. Generally, the effect of these changes might compromise several aspects of foods such as nutritional, microbiological and sensory quality. In the world, 32% of food is generally lost or wasted, as estimated by the Food and Agriculture Organization of the United (FAO). Global quantitative food losses and waste per year are roughly 30% for cereals, 40–50% for root crops, fruits and vegetables, 20% for oil seeds, meat and dairy and35% for fish [2]. Moreover, throwing food away implicates economic and environmental issues [3]. Thus, the possibility of using these wastes to obtain ingredients for the food industry could represent a significant step towards maintaining environmental balance.

During the last years, technological developments such as new packaging materials have been developed for food preservation to meet growing consumer demands of safe and perishable food products [4]. Modified atmosphere packaging (MAP) or synthetic antioxidants, such as butylated hydroxyanisole (BHA), butylated-hydroxytoluene (BHT), propyl gallate (PG), and *tert*-butylhydroquinone (TBHQ), represent efficient additives aimed at extending the shelf-life of foods. However, different studies revealed that these compounds are involved in many health issues, including cancer and carcinogenesis, causing a general consumer rejection of synthetic food additives [5,6]. Consequently, there is an increasing trend among food scientists to replace these synthetic preservatives with natural ones (herbal and spice extracts) which are supposed to be safer and may also add flavor [7]. Some studies have been carried out on natural antimicrobials, with a broad spectrum of activities, that can be used to extend the shelf-life of perishable foods [8]. The extracts from vegetable matrices, such as agri-food chain wastes and by-products, rich in polyphenols have been reported as good substitutes to extend the shelf-life of different food products [9]. 

Olive leaves are produced in high amounts in the olive oil industries (10% of the total weight of the olives) and accumulate during pruning of olive trees [10]. The most abundant phenolic component of olive leaves extract (OLE) is oleuropein [11], which exerts an in vitro inhibitory effect against many foodborne pathogens such as *Staphylococcus aureus*, *E. coli*, *Salmonella* spp., and *L. monocytogenes* and could be suitable for use in the food industry as natural preservatives [12]. In fact, several studies have reported the application of OLE to retard refined oil oxidative rancidity [13,14,15]; to increase the nutritional value of table olives [16,17]; to enhance the quality and shelf-life of cooked meat products [18,19], fresh seafood products [20] and bakery products [21].

The scientific literature contains no information regarding the use of OLE as a functional ingredient to extend the shelf-life of ready-to-use vegetable products, or as antimicrobial extract in the preparation and storage of minimally or non-thermally treated vegetable products. It is well known that the thermal treatment causes a reduction of the quality of foods in terms of colour, nutritive value and flavor [22]. 

Among non-thermally stabilized products, olive-based pâté is a gourmet product of Mediterranean tradition whose characteristics essentially depends both on the quality level of major ingredients used in their production and on technological process [23]. This product is usually stored in MAP at refrigerated conditions. In this framework, the aim of this work was to evaluate the effect of olive leaf extract addition on physico-chemical, microbiological and sensory aspects of non-thermally stabilized olive-based pâté during storage, making a comparison with the synthetic antioxidant BHT.

## 2. Materials and Methods

### 2.1. Raw Materials

Greek method fermented table olives (cv. Termite di Bitetto) were provided by a local farm in Modugno (Apulia, Italy); extra-virgin olive oil (cv. Coratina) came from an olive oil manufacturer in Andria (Apulia, Italy). Olive leaves (*Olea europaea* L., cultivar Coratina) were collected, processed and the polyphenols extraction was carried out according to the method described by Difonzo et al. [12]. and it had a total phenol content of 190 mg g^−1^ of dry matter and an antioxidant activity of about 738 µmol Trolox equivalents (TE) dry weight g^−1^.

### 2.2. Chemicals

Sodium carbonate and hexane were purchased from J.T. Baker (Deventer, Holland) and Carlo Erba (Milan, Italy), respectively; 1-propanol, butylhydroxytoluene (BHT), methanol, acetic acid, Folin–Ciocalteu reagent (2 N), gallic acid, ABTS (2,20-azino-bis(3-ethylbenzothiazoline-6-sulphonic acid) diammonium salt), DPPH (2,2-diphenyl-1-picrylhydrazyl), and Trolox ((±)-6-hydroxy-2,5,7,8-tetramethylchromane-2-carboxylic acid) were purchased from Sigma Aldrich (Milan, Italy). 

A modified argon-based atmosphere was used (Ar:CO_2_:H_2_; 75:23:2). Gas mixture was provided by Rivoira S.p.A. (Modugno, Apulia, Italy). A plastic film composed of orientated polyamide/polypropylene (OPA/PP) (Orved, Veneto, Italy). The gas permeability of packaging film was ≥27 ± 0.70 cm^3^ m^−2^ 24 h^−1^ for O_2_, and ≥120 ± 1 cm^3^ m^−2^ 24 h^−1^ for CO_2_.

### 2.3. Formulation and Manufacture of Olive-Based Pâté

Olive-based pâté was produced with 840 g kg^−1^ of fermented table olives and 160 g kg^−1^ of extra-virgin olive oil. All ingredients were mixed using a homogenizer (WFP16SE, Waring Commercial, Torrington, CT, USA) for 5 min to produce a homogeneous creamy pâté. Four kinds of olive-based pâté were produced: (i) control olive pâté without any supplementary antioxidant (CTR); (ii) OLE added at the concentration of 0.5 g kg^−1^ (EX0.5); (iii) OLE added at the concentration of 1.0 g kg^−1^(EX1); (iv) BHT at the concentration of 0.2 g kg^−1^ (BHT). After homogenization, approximately 70 g of each mixture was transferred into plastic trays (95 × 10 mm), and a stainless-steel heat sealer (VGP 25n, Orved, Musile di Piave, Veneto, Italy) was used to pack under argon-based atmosphere for a total of 96 sample. After packaging, the products were stored at 4 °C, and then sampled after 1, 15, 30, 45, 60, 75, 90 and 120 days of storage (T1, T15, T30, T45, T60, T75, T90, and T120). Three independent production trials were carried out for each sampling time and for each batch.

### 2.4. Microbiological Analyses

Microbiological analyses were carried out as reported by Cosmai et al. [24] and Caponio et al. (2019) [17]. Accordingly, 5 g of each olive based pate samples, added to 45 mL of 0.9% NaCl solution, were homogenized with a Stomacher Lab-Blender 400 (Seward Medical, London, United Kingdom) for 3 min. Cell densities of presumptive bacteria, yeasts and moulds were determined by plating serially diluted samples in different agar media: Plate count agar (total mesophilic aerobes); Wilkins–Chalgren anaerobe agar (total mesophilic anaerobes); MRS agar supplemented with cycloheximide at 0.17 g/L (lactic acid bacteria); M17 plus glucose 20% (*v*/*v*) (lactic streptococci); Slanetz and Bartley (enterococci); Violet Red Bile Glucose Agar (*Enterobacteriaceae*); GSP agar (Fluka, St. Louis, MO, USA) plus penicillin-G (60 g L^−1^) (*Pseudomonas* spp.); Listeria Selective Agar plates supplemented with modified Listeria Selective Supplement (*Listeria monocytogenes*). The number of yeasts and moulds were estimated on Wort Agar supplemented with chloramphenicol (0.1 g L^−1^). Spread-plating method was used to enumerate viable bacterial colonies of total mesophilic anaerobes, enterococci, *Pseudomonas* spp., and *Listeria monocytogenes*; pour-plating for the other ones. All media were purchased by Oxoid Ltd (Hampshire, UK) except for GSP agar. 

### 2.5. Physico-Chemical Analyses

The measurement of pH was carried out according to Cosmai et al. (2017) [24]. Headspace gas samples were withdrawn with a gas-tight syringe and the concentrations of O_2_ and CO_2_ were monitored with O_2_ and CO_2_ gas analyzers (CheckMate 3, DanSensor, Ametek, Milan).

Colourimetric readings were taken using a Minolta Chroma meter CR-300 (Osaka, Japan) with a CR 300 measurement head and CIE Standard Illuminant D65. The analysis was performed by placing the sample in a transparent quartz container. Lightness (*L**), redness (*a**, ±red-green), and yellowness (*b**, ±yellow-blue) were determined as colour coordinates.

For volatile compounds determination, the sample (1.00 ± 0.05 g) was weighed into 20-mL vials, sealed with a screw top aluminium cap and pierceable butyl rubber septa, and submitted to the SPME/GC-MS as reported in Caponio et al. [25]. 

### 2.6. Sensory Analysis

A trained panel of ten judges performed the sensory assessment. Quality attributes of colour (green colour) and odor (olive flavour) were evaluated using a 9-box scale labelled on the top with “dislike very much”, in the middle “neither like nor dislike” and on the bottom “like very much”, as also reported by Fabiano et al. [26]. Random samples were evaluated in individual plastic cups. 

### 2.7. Statistical Analysis

All the analytical determinations were carried on three independent batches for each sampling time and type of antioxidant used. Analysis of variance (one-way ANOVA) and the Tukey’s test were performed; the significant differences among the values were determined at *p* ≤ 0.05. All data were processed by the XLStat software (Addinsoft SARL, New York, NY, USA). Permutation analysis was also performed for the microbiological data using PermutMatrix [27].

## 3. Results and Discussion

The O_2_ concentration in the packages was close to 0% at the beginning of the experiment, and increased to a maximum of 0.3% at the end of storage. This data confirms that the adopted packaging maintained low O_2_ concentrations and limited its increase during storage. It is known that different factors can influence the residue of the O_2_ in the headspace of the samples, such as (i) the effectiveness of the MAP equipment in removing air and (ii) the permeability of the tray-wrap packaging materials in enabling the oxygen exchanges from the headspace of the tray towards the master bag headspace [28]. Moreover, the CO_2_ concentration inside packages showed a slight decrease, as observed in other studies [29,30].

Changes in colour parameters (*L**, *a** and *b** values) of olive-based pâté during storage are given in Figure 1. The value of *L** decreased during storage whereas the colorimetric index *a** increased. This result is in agreement with Lorenzo et al. [9] and Cosmai et al. [22] and could be due to moisture losses or pigment oxidation. The yellowness index (*b**) showed an undefined trend. In some cases, *L** was significantly lower for the samples added with OLE than BHT and/or CTR; the opposite trend, instead, appeared for the index *a** even if in the most of cases no significant differences between the samples were appreciable. 

Figure 2 and Table 1 shows the growth of microorganisms in olive-based pâté under study. Overall, pathogens (*Clostridium* and *Listeria* spp.) and contaminants (*Pseudomonas* spp. and *Enterobacteriaceae*) were not found. The samples of olive-based pâté were not subjected to thermal stabilization treatment and, as expected, cultivable bacteria, yeasts and moulds were detected during samples production and storage. The microbiota occurring was mainly affected by the addition of OLE and refrigeration storage. The main microbial groups registered a significant loss of ca. 0.5–1 logarithmic cycles in samples added with extract, especially with 1.0 g kg^−1^ (EX1). The addition of BHT or OLE at the concentration of 0.5 g kg^−1^ (EX0.5) did not modify significantly the microbiota. The permutation analysis based on cell density (Figure 3) distributed the samples into eight major clusters (A-H). The CTR and BHT added samples almost grouped in the same clusters, being the OLE-added olive-based samples that most diverse, especially with 1.0 g kg^−1^ (EX1). According to Cosmai et al. [24], the cell density of the monitored microorganisms (Table 1) decreased significantly (*p* < 0.05) in all samples throughout storage period, but the addition of OLE significantly (*p* < 0.05) decreased the microbial number in treated samples. This might be due to the direct inhibitory action of OLE on microbial growth and presence of polyphenolic compounds in OLE possessing anti-microbial properties. The yeast and mould counts were detected during all the storage in CTR and BHT conditions. Significantly (*p* < 0.05) lower counts were recorded for treated olive-based pâté that progressively disappeared. This could be due to the presence of polyphenolic compounds in OLE exerting antifungal and antimicrobial properties [31]. Thus, OLE can be effectively utilized in the development of olive-based pâté with enhanced oxidative stability and microbial quality. The samples were recorded fit for consumption, by a microbiological point of view, during refrigeration (4 ± 1 °C) storage for 120 days in MAP conditions, as compared to the 90 days of the untreated or BHT added samples.

Under the experimental conditions of this study, the pH was maintained below 4.5 in all samples, similar to those observed by Cosmai et al. [24], and no statistical differences (*p* > 0.05) were observed among the samples during storage (Figure 3). The pH of the olive-based pâté is a crucial parameter from the sanitary point of view. The observed values could explain the absence of pathogens and spoilage microorganisms during sample production and storage.

Thielmann et al. [32] reviewed the antimicrobial activity of olive leaf against a wide variety of bacteria and fungi. OLE contains polyphenols that exhibit growth inhibitory and bactericidal potential against various Gram-positive and -negative bacteria. In agreement with the result of our study, Gok and Bor [33] showed an influence of their OLE on the microbiological parameters considered in fresh beef. In comparison to untreated references the applied OLE concentrations delayed microbial growth about 1.95 and 2.27 log CFU g^−1^, respectively, in a 10 days’ storage test at 7 °C. Moreover, the sensory characteristics improved by adding OLE. Hayes et al. [34] on the contrary, reported no growth inhibitory potential for OLE in fresh beef preparations. 

The results of the sensory evaluation performed on olive-based pâté during storage was aimed at assessing the influence of OLE addition on the olive-based pâté colour and odor appreciation on a scale from 0 to 10. Figure 4 clearly showed a decline in odour and colour acceptability at T90. At T1 no significant colour differences between the samples added with OLE and CTR were detected; the odour appreciation was significantly higher in the samples EX1 than in samples added with BHT. At T90 both odour and colour were most appreciated from panelists in the samples EX1 that was significantly different than BHT. Moreover, the panelists perceived no sensory defects for all the samples, even if a greater typical olive flavour was perceived in samples added with EX than in BHT and CTR samples. This result was confirmed by a significantly higher content of 2-methylbutanal, 3-methylbutanal, and ethyl alcohol in samples added with EX, than CTR and BHT samples (Table 2). In fact, according to Cosmai et al. [22], 2-methylbutanal and 3-methylbutanal are among the most involved compounds in olive flavour. After 90 days of storage, the samples CTR and BHT showed evident sensorial defects and were not acceptable for human consumption. 

Table 2 shows the amount of volatile compounds at T90 (close to the end of shelf-life, as confirmed by the microbiological and sensorial analyses). The addition of OLE to olive based-pâté produced significant increases above all in aldehydes, ketones, and esters. Not significant differences were observed in carboxylic acids and alcohols except for hexanoic acid and ethyl alcohol, respectively. Ethanol, end-product of the Embden-Meyerhof-Parnas glycolytic pathway, is an important flavour component. The biosynthesis of higher alcohols is considered to be linked to the deamination of amino acids [35]. Moreover, acetic acid is produced by bacteria and yeasts by oxidation of ethanol. Propionic acid is produced by bacteria *Propionibacterium* species as the product of breakdown of fatty acids and/or amino acids. The volatile compounds detected in our samples showed that heterolactic and propionic fermentation occurred. The same trend was obtained for ethyl acetate ester. It is well known that acetate esters are synthesized by the enzyme alcohol-acyl-transferase, which catalyses the esterification of volatile alcohols with acetyl CoA [36]. Moreover, 13-hydroperoxides of linoleic acid are split by hydroperoxide lyses to produce hexanal while other aldehydes such as octanal, nonanal and *trans*-2-heptenal are originated from lipid oxidation [37]. However, Raitio et al. [38] highlighted the interrelations of lipid oxidation and protein degradation in vegetable bouillon pâté, highlighting a correlation between the formation of hexanal and 3-methylbutanal. In fact, our results showed, in most of cases, that both 2 and 3-methylbutanal and hexanal were higher in the samples added with OLE. 

## 4. Conclusions

The results of sensory evaluation of olive-based pâté showed that no sensory defects for all the samples were perceived, even if a greater typical olive flavour was present in samples added with OLE than in BHT and CTR. These results were confirmed by a significantly higher level of 2-methylbutanal and 3-methylbutanal in the samples added with OLE than CTR and BHT. Moreover, the main microbial groups registered a significant loss of 0.5–1 logarithmic cycles in samples added of OLE, especially when it was used at the highest concentration (1.0 g kg^−1^). To conclude, the results of the present study indicate the high potential of using OLE as natural preservative in non-thermally stabilized olive-based pâté, since some spoilage-related microbial groups were negatively affected by the addition of OLE at the highest concentration.

## Figures and Tables

**Figure 1 foods-08-00138-f001:**
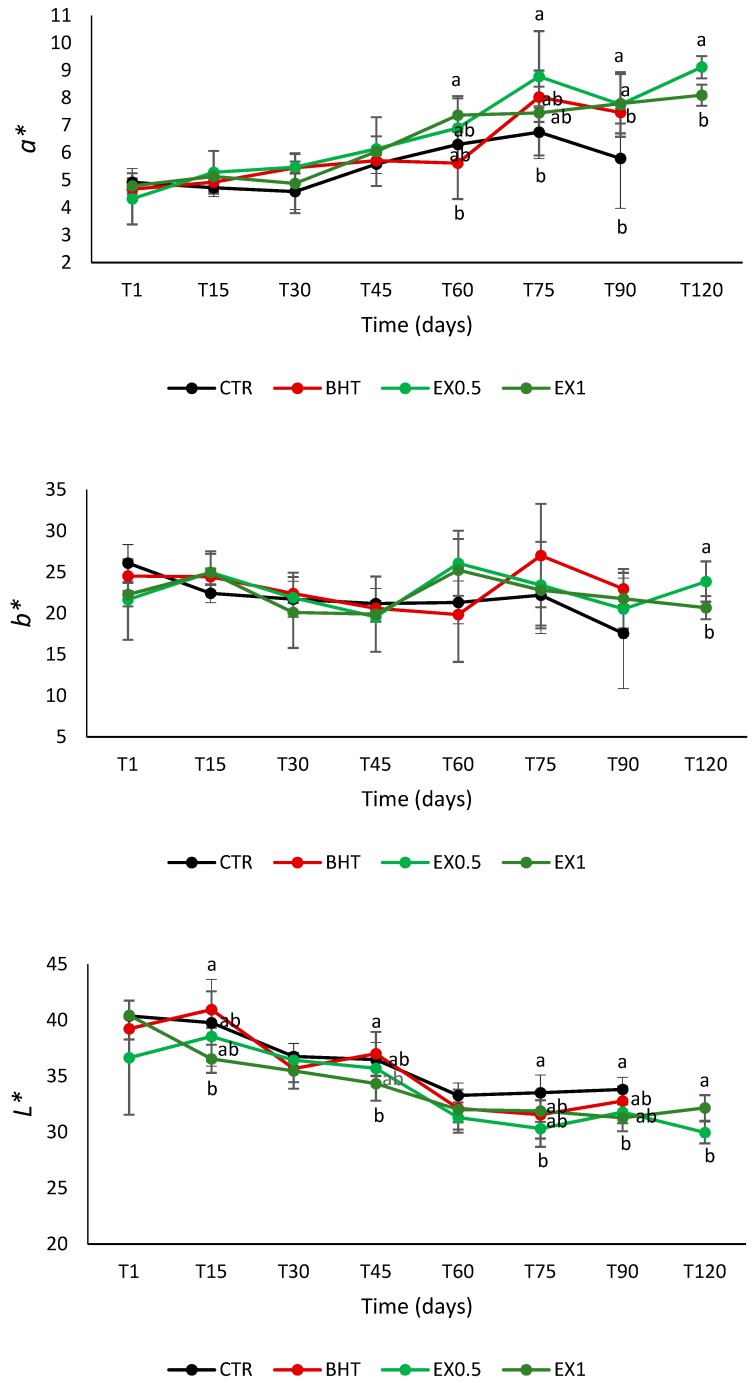
Mean values ± standard deviation of colorimetric parameters (*a**, *b**, *L**). CTR: control samples: BHT: samples added with the synthetic antioxidant; EX0.5: samples added with olive leaf extract at 0.5 g kg^−1^; EX1: samples added with olive leaf extract at 1 g kg^−1^. One-way ANOVA and Tukey’s test were performed among the samples for each time. Different letters indicate significant differences for *p* < 0.05.

**Figure 2 foods-08-00138-f002:**
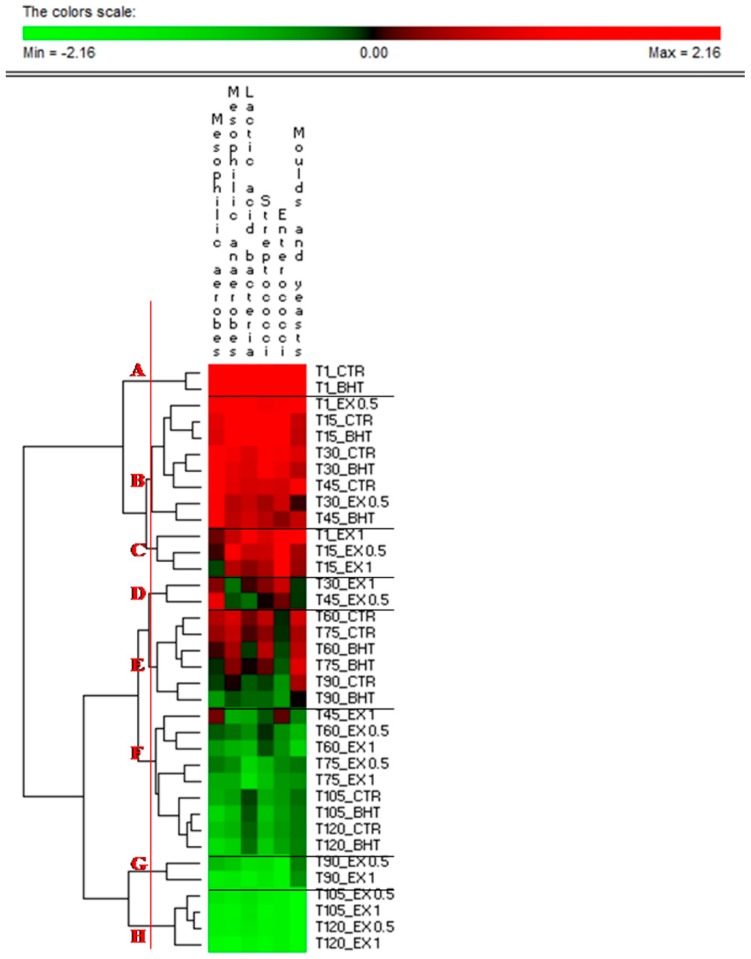
Permutation analysis of cell density during storage of olive-based paste. Euclidean distance and McQuitty’s criterion (weighted pair group method with averages) were used for clustering. The colours correspond to normalized mean data levels from low (green) to high (red). The colour scale, in terms of units of standard deviation, is shown at the top. Eight clusters are identified by the letters A-H. CTR: control samples; BHT: olive-based paste added of butylated hydroxytoluene; EX0.5: olive-based paste added of 0.5 g kg^−1^ of olive leaves extract; EX1: olive-based paste added of 1 g kg^−1^ of olive leaves extract; T1, T15, T30, T45, T60, T75, T90, T105, and T120: 1, 15, 30, 45, 60, 75, 90, 105 and 120 storage days.

**Figure 3 foods-08-00138-f003:**
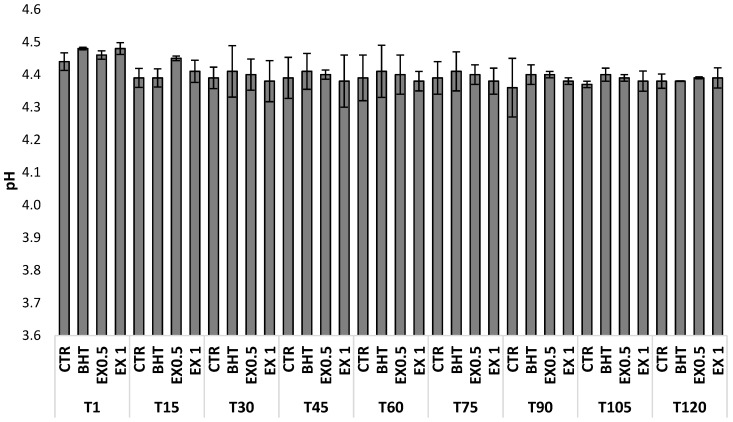
Trend of the pH variations during storage of olive-based paste. CTR: control samples; BHT: olive-based paste added of butylated hydroxytoluene; EX0.5: olive-based paste added of 0.5 g kg^−1^ of olive leaves extract; EX1: olive-based paste added of 1 g kg^−1^ of olive leaves extract; T1, T15, T30, T45, T60, T75, T90, T105, and T120: 1, 15, 30, 45, 60, 75, 90, 105 and 120 storage days. Data are the means from three independent experiments. Bars represent standard deviations.

**Figure 4 foods-08-00138-f004:**
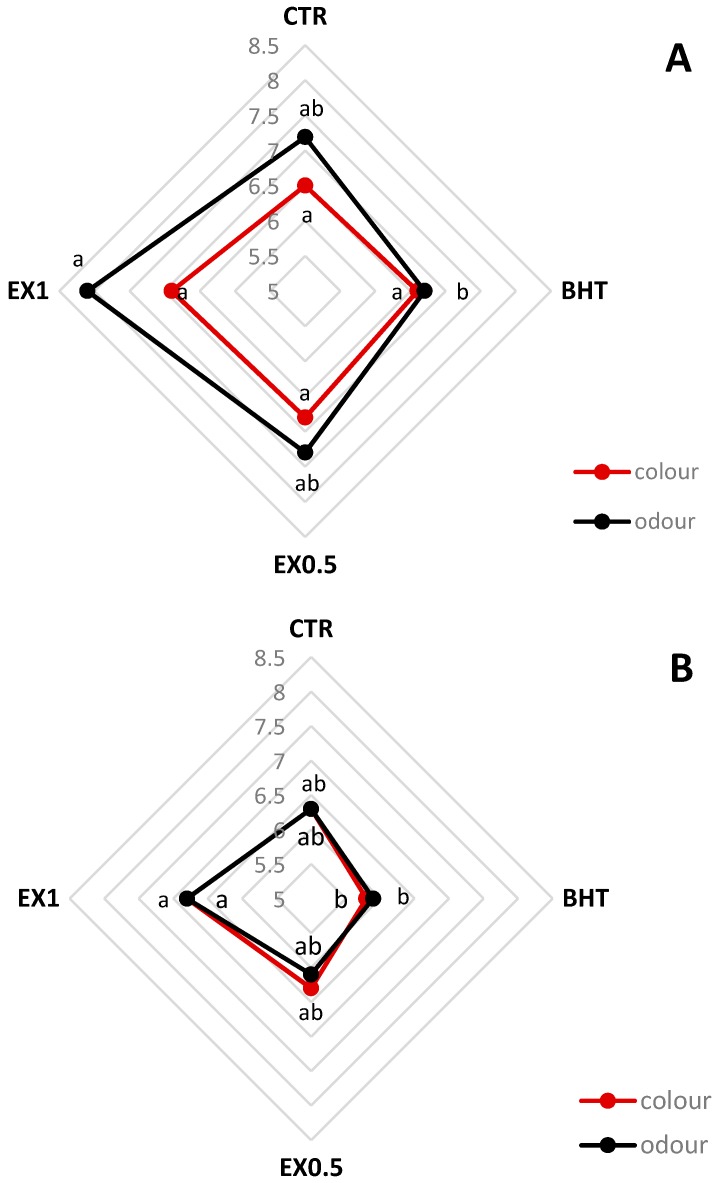
Results of sensory evaluation at T1 (**A**) and T90 (**B**). CTR, control samples; BHT, samples added with the synthetic antioxidant; EX0.5, samples added with olive leaf extract at 0.5 g kg^−1^; EX1, samples added with olive leaf extract at 1 g kg^−1^. One-way ANOVA and Tukey’s test were performed among the samples for each time. Different letters indicate significant differences for *p* < 0.05.

**Table 1 foods-08-00138-t001:** Cell densities (log CFU g^−1^) and standard deviation of microbial groups detected in olive-based paste during storage time.

Time	Samples Code	Mesophilic Aerobes	Mesophilic Anaerobes	Lactic Acid Bacteria	Streptococci	Enterococci	Moulds and Yeasts
T0	CTR	5.98 ± 0.15 a	6.04 ± 0.27 a	6.55 ± 0.14 a	6.19 ± 0.14 a	5.15 ± 0.13 ab	5.26 ± 0.27 a
	BHT	5.92 ± 0.24 a	5.88 ± 0.14 ab	6.49 ± 0.19 b	6.16 ± 0.12 a	5.28 ± 0.27 a	5.19 ± 0.14 a
	EX0.5	5.82 ± 0.23 a	5.81 ± 0.15 ab	6.07 ± 0.28 b	5.76 ± 0.16 b	4.82 ± 0.26 bc	4.98 ± 0.22 b
	EX1	5.40 ± 0.16 b	5.57 ± 0.15 ab	5.94 ± 0.15 b	5.56 ± 0.18 bc	4.83 ± 0.22 bc	5.06 ± 0.10 b
T15	CTR	5.71 ± 0.13 ab	5.89 ± 0.26 b	6.11 ± 0.26 a	5.96 ± 0.15 ab	5.27 ± 0.19 a	4.05 ± 0.15 d
	BHT	5.68 ± 0.08 ab	5.84 ± 0.11 ab	6.05 ± 0.13 b	6.04 ± 0.13 ab	5.16 ± 0.11 ab	3.99 ± 0.16 d
	EX0.5	5.35 ± 0.12 bc	5.78 ± 0.13 b	5.67 ± 0.12 c	5.47 ± 0.14 c	4.98 ± 0.11 b	3.74 ± 0.16 de
	EX1	5.26 ± 0.18 bc	5.52 ± 0.27 b	5.46 ± 0.13 cd	5.31 ± 0.15 c	4.99 ± 0.21 b	3.66 ± 0.09 e
T30	CTR	6.01 ± 0.13 a	5.74 ± 0.14 b	5.78 ± 0.11 c	5.81 ± 0.13 b	4.76 ± 0.1 bc	4.34 ± 0.19 c
	BHT	5.92 ± 0.16 a	5.68 ± 0.28 b	5.76 ± 0.02 c	5.88 ± 0.09 b	4.51 ± 0.11 c	3.91 ± 0.13 d
	EX0.5	5.82 ± 0.11 a	5.52 ± 0.13 b	5.67 ± 0.06 c	5.27 ± 0.3 cd	4.04 ± 0.09 d	3.22 ± 0.07 f
	EX1	5.44 ± 0.21 b	5.18 ± 0.11 c	5.33 ± 0.01 cd	5.17 ± 0.5 cd	3.99 ± 0.08 d	3.02 ± 0.06 fg
T45	CTR	5.91 ± 0.03 a	5.67 ± 0.13 b	5.71 ± 0.27 c	5.55 ± 0.13 bc	4.08 ± 0.09 d	4.85 ± 0.17 b
	BHT	5.83 ± 0.15 a	5.54 ± 0.27 b	5.72 ± 0.13 c	5.35 ± 0.12 c	3.35 ± 0.06 e	4.02 ± 0.09 d
	EX0.5	5.74 ± 0.02 ab	5.24 ± 0.02 c	5.12 ± 0.25 d	4.96 ± 0.09 de	3.17 ± 0.07 ef	3.03 ± 0.06 fg
	EX1	5.41 ± 0.26 b	5.03 ± 0.08 cd	4.92 ± 0.13 de	4.82 ± 0.08 de	3.08 ± 0.11 f	2.61 ± 0.06 g
T60	CTR	5.55 ± 0.24 b	5.62 ± 0.12 b	5.41 ± 0.08 cd	5.41 ± 0.12 c	2.75 ± 0.14 fg	4.21 ± 0.09 c
	BHT	5.35 ± 0.05 bc	5.48 ± 0.06 bc	5.23 ± 0.07 d	5.37 ± 0.13 c	2.61 ± 0.12 g	3.98 ± 0.09 d
	EX0.5	5.23 ± 0.23 bc	5.18 ± 0.11 c	5.02 ± 0.12 de	4.91 ± 0.12 de	2.08 ± 0.09 h	2.04 ± 0.04 i
	EX1	5.09 ± 0.17 c	5.01 ± 0.11 cd	4.82 ± 0.06 de	4.84 ± 0.18 de	2.02 ± 0.04 h	1.60 ± 0.08 l
T75	CTR	5.48 ± 0.06 b	5.56 ± 0.13 b	5.34 ± 0.25 cd	5.19 ± 0.12 cd	2.71 ± 0.06 fg	3.80 ± 0.11 de
	BHT	5.29 ± 0.01 bc	5.42 ± 0.23 bc	5.29 ± 0.09 d	5.08 ± 0.12 d	2.51 ± 0.05 g	4.36 ± 0.07 c
	EX0.5	5.17 ± 0.11 bc	5.12 ± 0.08 cd	4.63 ± 0.13 e	4.37 ± 0.11 ef	2.11 ± 0.04 h	2.68 ± 0.01 g
	EX1	5.03 ± 0.22 c	5.04 ± 0.01 cd	4.51 ± 0.14 e	4.32 ± 0.05 ef	1.96 ± 0.08 h	2.49 ± 0.09 h
T90	CTR	5.27 ± 0.09 bc	5.31 ± 0.01 c	5.17 ± 0.01 d	4.88 ± 0.13 de	2.01 ± 0.20 h	3.77 ± 0.08 de
	BHT	5.08 ± 0.01 c	5.24 ± 0.09 c	5.14 ± 0.09 d	4.77 ± 0.12 e	1.96 ± 0.15 h	3.14 ± 0.07 f
	EX0.5	4.96 ± 0.11 c	4.91 ± 0.11 d	4.52 ± 0.12 e	4.01 ± 0.08 fg	<1l **	2.62 ± 0.02 g
	EX1	4.63 ± 0.08 d	4.73 ± 0.21 de	4.31 ± 0.11 f	3.92 ± 0.11 fg	<1l	2.45 ± 0.04 h
T105	CTR	5.01 ± 0.11 c	5.07 ± 0.16 cd	5.22 ± 0.11 d	4.46 ± 0.13 e	1.95 ± 0.04 h	2.74 ± 0.06 g
	BHT	4.83 ± 0.09 cd	4.96 ± 0.14 d	5.18 ± 0.08 d	4.28 ± 0.14 f	1.87 ± 0.03 i	2.66 g
	EX0.5	4.72 ± 0.22 cd	4.69 ± 0.10 de	4.52 ± 0.03 e	3.74 ± 0.09 g	<1l	<1 m
	EX1	4.61 ± 0.07 d	4.58 ± 0.10 e	4.47 ± 0.05e f	3.61 ± 0.08 g	<1l	<1 m
T120	CTR	4.92 ± 0.02 c	5.00 ± 0.00 cd	5.15 ± 0.02 d	4.38 ± 0.05 ef	1.93 h	2.65 ± 0.026 g
	BHT	4.77 ± 0.02 cd	4.88 ± 0.02 d	5.12 ± 0.05 d	4.22 ± 0.09 f	1.81 i	2.61 ± 0.044 g
	EX0.5	4.61 ± 0.01 d	4.63 ± 0.02 de	4.43 ± 0.02 ef	3.68 ± 0.07 g	<1l	<1 m
	EX1	4.48 ± 0.01 e	4.46 ± 0.02 e	4.29 ± 0.00 f	3.55 ± 0.04 g	<1l	<1 m

The values represent means of duplicates; means followed by different letters in the same column are significantly different (*p* < 0.05) according to Tukey’s HSD test. ** <1 log CFU g^−1^. CTR: control samples; BHT: olive-based paste added of butylated hydroxytoluene. EX0.5: olive-based paste added of 0.5 g kg^−1^ of olive leaves extract; EX1: olive-based paste added of 1 g kg^−1^ of olive leaves extract; T1, T15, T30, T45, T60, T75, T90, T105, and T120: 1, 15, 30, 45, 60, 75, 90, 105 and 120 storage days.

**Table 2 foods-08-00138-t002:** Mean values, standard deviation and results of one-way ANOVA performed on volatile compounds detected in samples after 90 days of storage.

Volatile Compounds	CTR	BHT	EX0.5	EX1
Aldehydes and ketones				
2-Propenal	10.31 ± 1.55 b	20.17 ± 0.78 a	15.69 ± 3.46 ab	21.12 ± 5.06 a
2-Methylbutanal	3.73 ± 0.63 c	8.20 ± 0.36 b	10.64 ± 0.61 a	12.07 ± 1.17 a
3-Methylbutanal	6.18 ± 1.30 b	15.03 ± 1.43 ab	19.27 ± 0.89 ab	23.21 ± 2.27 a
Hexanal	2.12 ± 0.22 b	3.76 ± 0.75 ab	6.18 ± 0.41 ab	6.58 ± 1.21 a
2-(*E*)-Hexenal	0.65 ± 0.18 ab	0.38 ± 0.04 b	0.94 ± 0.54 a	0.57 ± 0.07 ab
2-(*E*)-Heptenal	0.89 ± 0.02 c	1.59 ± 0.37 b	3.59 ± 1.18 a	3.29 ± 0.36 a
Nonanal	1.06 ± 0.06 a	0.75 ± 0.14 b	1.01 ± 0.08 a	0.88 ± 0.08 a
2-Butanone	530.92 ± 21.19	568.68 ± 24.63	555.29 ± 28.55	593.65 ± 23.08
Benzaldehyde	1.22 ± 0.42	1.29 ± 0.27	1.39 ± 0.14	1.25 ± 0.45
Benzeneacetaldehyde	2.30 ± 1.66	2.33 ± 0.84	2.42 ± 0.79	4.78 ± 0.68
Esters				
Ethyl acetate	62.26 ± 6.43 b	70.69 ± 2.79 b	111.78 ± 10.80 a	110.12 ± 2.05 a
n-Propyl acetate	30.40 ± 7.50	42.19 ± 6.58	45.37 ± 4.33	41.46 ± 7.19
Alcohols				
Ethyl alcohol	96.65 ± 1.35 b	117.50 ± 9.39 b	186.76 ± 17.02 a	193.77 ± 22.77 a
2-Butanol	252.27 ± 6.12	248.71 ± 13.48	236.95 ± 30.81	218.16 ± 9.33
Acids				
Acetic acid	249.85 ± 79.21	186.99 ± 52.01	249.20 ± 33.23	240.28 ± 35.91
Propanoic acid	29.74 ± 7.89	19.87 ± 1.32	31.01 ± 1.24	22.91 ± 0.89
Butanoic acid	0.77 ± 0.10 b	0.98 ± 0.18 ab	1.29 ± 0.16 a	1.05 ± 0.11 ab
Hexanoic acid	2.88 ± 0.09 a	2.78 ± 0.10 a	1.60 ± 0.24 b	1.24 ± 0.05 b

The results are expressed as µg g^−1^. Different letters in the same line are significantly different (*p* < 0.05) according to Tukey’s test. CTR: control samples; BHT: olive-based paste added of butylated hydroxytoluene; EX0.5: olive-based paste added of 0.5 g kg^−1^ of olive leaves extract; EX1: olive-based paste added of 1 g kg^-1^ of olive leaves extract.

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
