# Peer review of "Physico-Chemical, Microbiological and Sensory Evaluation of Ready-to-Use Vegetable Pâté Added with Olive Leaf Extract"

_foods, 2019, doi:10.3390/foods8040138_

Reviewer 1 Report

The manuscript aimed at investigating the effects of crude olive leaf extract as a supplement to olive-based paste. The authors should to be more careful about concluding remarks (specially in the abstract), e.g., this study used olive paste as food product and information on the usage of olive extract in olive-based pastes does not have much value on other non-thermally stabilized foods.

Some comments are as follows:

From the title, any reader will expect information regarding “increase of shelf-life” of food products, but the manuscript did not present much information about it. The tile should re-worded to better represent experimental findings.

The authors used crude extract of olive leaves. Crude phenolic extracts tend to show highly variable effect (as their composition varies based on type of olives, season, among other factors). Did the authors carry out the experiment with OLE from only one batch of olive leaves?

What are the common preservatives that are used in commercial pâté? The authors need to justify why OLE was compared with BHT.

Statistical analysis: Did the authors carry out any post-hoc test for multiple mean comparison?

Author Response

Dear Reviewers,

thank you very much for your careful revision and helpful suggestions.

In agreement with your observations, the manuscript has been carefully revised and the following modifications have been introduced.

Sincerely,

Francesco Caponio

Reviewer #1:

Comments and Suggestions for Authors

The manuscript aimed at investigating the effects of crude olive leaf extract as a supplement to olive-based paste. The authors should to be more careful about concluding remarks (specially in the abstract), e.g., this study used olive paste as food product and information on the usage of olive extract in olive-based pastes does not have much value on other non-thermally stabilized foods.

We added an additional period in the abstract to better explain the results, according to your observation.

Some comments are as follows:

From the title, any reader will expect information regarding “increase of shelf-life” of food products, but the manuscript did not present much information about it. The tile should re-worded to better represent experimental findings.

Thank you for your suggestion, we have modified the title.

The authors used crude extract of olive leaves. Crude phenolic extracts tend to show highly variable effect (as their composition varies based on type of olives, season, among other factors). Did the authors carry out the experiment with OLE from only one batch of olive leaves?

Thank you for this comment. One of the main item of raw extract from wastes is the lack of standardization. However, we used specifically a raw extract from cv. Coratina leaves; in our laboratory we have produced extract in different season and after HPLC characterization of phenolic compounds and quantitation, the differences between the season is not substantial. Anyway, we’ve added information about antioxidant activity value and total phenol content in the paragraph “Raw materials” (lines 91-92).

What are the common preservatives that are used in commercial pâté? The authors need to justify why OLE was compared with BHT.

BHT represent one of the main additives used in foods. Several authors added BHT as additive in patè (Pinho, Ferreira and Oliveira, 2000; Estevez, Ventana, Ramirez and Ramon, 2004). Moreover, this topic was explained in the Introduction (lines 48-55).

Statistical analysis: Did the authors carry out any post-hoc test for multiple mean comparison?

Yes, the Tukey’s Test was performed. Please see the figures and tables caption. Moreover, we added this information in the paragraph “Statistical analysis”.

Reviewer 2 Report

As requested, I reviewed the manuscript entitled ‘Olive leaf extract ……vegetable pate’. This manuscript needs a major revision. First, I would suggest authors to change the ‘Title’ as olive leaf extract is not a tool but a natural preservative. There are issues in English language and sentence structure throughout the manuscript that needs major revision.

Other comments:

-Please explain the results part in order as shown in methods. First explain physicochemical, then move to microbiological and at the end discuss sensory evaluation. Methods should match the results & discussion section.

-Please include the sample size and method of plating in microbiological part.

-Table 1: Authors indicated < 1 log for < 11 CFU. It should be < 10 CFU. In addition, what < 1 m stands for Yeasts & Molds count?

-References style in text are not correct. Please revise.

-Please see the attached pdf for the detail comments and suggestions.

Author Response

Dear Reviewers,

thank you very much for your careful revision and helpful suggestions.

In agreement with your observations, the manuscript has been carefully revised and the following modifications have been introduced.

Sincerely,

Francesco Caponio

Reviewer #2:

Comments and Suggestions for Authors

As requested, I reviewed the manuscript entitled ‘Olive leaf extract ……vegetable pate’. This manuscript needs a major revision. First, I would suggest authors to change the ‘Title’ as olive leaf extract is not a tool but a natural preservative. There are issues in English language and sentence structure throughout the manuscript that needs major revision.

Tank you for your suggestions. We have modified the title and the English language was revised throughout the whole manuscript.

Other comments:

-Please explain the results part in order as shown in methods. First explain physicochemical, then move to microbiological and at the end discuss sensory evaluation. Methods should match the results & discussion section.

We modified the sequence of the results according to your suggestion. We moved the physico-chemical after the microbiological analyses, and we finished with sensory evaluation and volatile compounds description since they are related with the discussed sensory aspects.

-Please include the sample size and method of plating in microbiological part.

According to your suggestion, the sample size and method of plating were added in the microbiological part.

-Table 1: Authors indicated < 1 log for < 11 CFU. It should be < 10 CFU. In addition, what < 1 m stands for Yeasts & Molds count?

Thank you,<1 log CFU g-1 indicated <10 CFU g-1, while <1 m for Yeasts & Moulds count indicated <10 CFU g-1 ; the letter indicated the result based on Tukey’s HSD test.

-References style in text are not correct. Please revise.

Sorry for the mistakes, we have  revised the reference style in the text.

-Please see the attached pdf for the detail comments and suggestions.

Thank you for all comments and suggestions in the text. We have revised all you have indicated in the pdf.

Round  2

Reviewer 2 Report

Authors have made the changes and address the concerned issues. Quality of manuscript has improved.